# In Vitro Evaluation of the Antimicrobial Activity of Eighteen Essential Oils Against Gram-Positive and Gram-Negative Bacteria in Two Different Growth Media

**DOI:** 10.3390/pathogens14121216

**Published:** 2025-11-29

**Authors:** Cinzia Marianelli, Sonia Ferraiuolo, Martina Topini, Laura Narciso

**Affiliations:** Department of Food Safety, Nutrition and Veterinary Public Health, Istituto Superiore di Sanità, 00161 Rome, Italy; soniaferraiuolo97@gmail.com (S.F.); martinatopini@gmail.com (M.T.); laura.narciso@iss.it (L.N.)

**Keywords:** essential oils, Gram-positive bacteria, Gram-negative bacteria, spot-on-agar test

## Abstract

The rise in antimicrobial resistance and tolerance over time represents a significant threat to human and animal health. This has led to a notable increase in interest within the scientific community in the development of herbal-based therapeutic alternatives to antibiotics. The present study aimed at evaluating the in vitro antimicrobial activity of 18 essential oils (EOs) against a total of 17 strains belonging to both Gram-positive and Gram-negative bacteria, by employing the spot-on-agar method using two different culture media, Brain Heart Infusion (BHI) and Mueller–Hinton (MH). The antimicrobial properties of these essential oils were investigated, as well as their antimicrobial stability over a period of seven days. The overall efficacy of the EOs remained consistent over seven days, regardless of the solid medium used. However, the antimicrobial effects of the EOs were more pronounced in BHI than in MH for Gram-positive bacteria. While variations in antimicrobial activity were recorded among different species and strains, oregano EO proved to be the most effective against both Gram-positive and Gram-negative bacteria, followed by thyme and tea tree. The findings of this study support the notion that EOs could be employed as a promising therapeutic strategy to combat antibiotic-resistant bacteria, thereby enhancing the efforts aimed at addressing antibiotic resistance. Furthermore, the divergent antimicrobial effects exhibited by the two growth media employed here may facilitate the establishment of standardised protocols for the testing of EOs against bacteria.

## 1. Introduction

Antimicrobial resistance (AMR) is a significant and growing public health threat on a global scale, with implications for all countries. The phenomenon is not limited to human pathogens; it also affects animal and plant pathogens, with consequences for food security and environmental health. The presence of antimicrobial-resistant pathogens poses significant challenges to treatment and control of infectious diseases, as these pathogens render standard treatment regimens ineffective [1]. Drivers of AMR are the overuse and misuse of antimicrobials over the last few decades, poverty, inadequate sanitation, poor infection control and lack of access to affordable medicines and vaccines [2].

In response to the global AMR threat, the World Health Organization (WHO) launched its first Bacterial Priority Pathogens List (BPPL) in 2017, which had the primary objective of guiding the research and development of new antibacterials. This list, which featured 25 antibiotic-resistant pathogens, was updated in May 2024 [3]. The 2024 BPPL introduced additional pathogens and reclassified others [4]. Some pathogens can escape antibiotic action due to their ability to acquire and transfer resistance via horizontal gene transfer and therefore grouped under the acronym ESKAPEE pathogens (*Enterococcus faecium*, *Staphylococcus aureus*, *Klebsiella pneumoniae*, *Acinetobacter baumannii*, *Pseudomonas aeruginosa*, *Enterobacter* spp., and *Escherichia coli*). The ESKAPEE pathogens are considered the major cause of life-threatening bacterial infections [5]. There is an urgent necessity for focused research, sustained investments and international collaboration in the field of novel antibacterials and vaccines. Furthermore, it is crucial to streamline surveillance and infection prevention and control measures in order to effectively address priority AMR pathogens [4].

As new antibiotics are only sporadically approved, natural antibacterial agents (derived from microorganisms, fungi, plants, animals and minerals) have seen a resurgence in interest as potential alternatives to conventional antibiotics and chemotherapeutics [6]. Plants, herbs, and their derivates are considered innovative therapeutic strategies for research focused on AMR [7]. Essential oils (EOs) are secondary metabolites produced by aromatic plants and rich sources of bioactive compounds with significant antimicrobial potential. EOs consist of a complex mixture of volatile and aromatic molecules, including terpenes, aldehydes, alcohols, ethers and phenols, which display intrinsic antibacterial properties and antibiotic resistance-modifying activities [8]. Several EOs, including thyme, oregano, cinnamon, lavender, sage, clove, mint and rosemary, have demonstrated strong antibacterial and antifungal activities, potentially through the disruption of cell wall integrity, the inhibition of enzymatic processes, the interference with metabolic pathways, the prevention of biofilm formation and the alteration of cell membrane potential [9,10,11]. Thymol, menthol and carvacrol from the monoterpene group, along with eugenol and cinnamaldehyde from the phenylpropanoids group, are among the most promising EO components with antibacterial activity [10]. Due to their complex composition, the development of resistance to these molecules is significantly limited [12]. The broad-spectrum antibacterial properties, in conjunction with the potential for low toxicity, render EOs and their components promising candidates for the management of infections and to overcoming AMR [7].

Until a standardised method is officially established for the assessment of the antimicrobial properties of EOs, researchers generally use methods adapted for the screening and evaluation of the antimicrobial activity of antibiotics. A number of in vitro testing methods for analyzing the antimicrobial activity of EOs have been documented in the extant literature [13]. The most common assays used for the determination of Minimum Inhibitory Concentrations (MICs) include broth dilution (where an EO is diluted at varying concentrations in a liquid medium) [13,14,15], and agar dilution (where an EO is mixed at varying concentrations with melted agar) methods [13]. Conversely, the conventional methodologies employed for the preliminary screening of the antimicrobial activities of EOs encompass agar diffusion techniques, which utilize paper disks impregnated with an EO and positioned on the agar surface [13,16], or wells (holes) drilled out into the agar and filled with an EO [13]. Alternatively, droplets of EOs are deposited directly onto the agar surface in a process also referred to as the spot-on-agar test [14,17,18]. If on one hand dilution methods are based on the determination of MIC values by growth/no growth end-point, on the other hand diffusion methods are based on the measurement of the size of the zone of inhibition [13,19]. The main disadvantage of both broth and agar dilution methods is that a large amount of EO and medium is required to dilute the EO to different concentrations [13]. In the agar dilution method, the agar must also be heated to 45–50 °C to keep it in a liquid state, before it is mixed with a defined amount of EO. This temperature can affect the physiochemical properties of EOs [13]. On the other hand, the main drawback of agar diffusion methods is that parameters such as agar composition and thickness can influence the size of the inhibition zone, resulting in less consistent outcomes compared to dilution assays. However, agar diffusion techniques are the preferred method for rapidly quantifying the antimicrobial activity of EOs [13]. In addition to the variety of testing methods employed, there is also a range of media used for the assessment of the antimicrobial activity of EOs [13]. Although the antimicrobial effects of EOs have been thoroughly studied in vitro over the past decades, the extant literature on EOs remains highly varied with regard to the methods employed, the growth media utilised, and the EOs examined. Consequently, the comparison of results is rendered more challenging. Further research is required to facilitate the establishment of a universally accepted methodology for the assessment of the antimicrobial properties of EOs and their compounds.

The present study aimed at evaluating the in vitro antimicrobial activity of 18 EOs against a range of Gram-positive and Gram-negative bacteria, including reference strains and opportunistic and clinical pathogens for a total of 17 strains, by employing the spot-on-agar. It is noteworthy that some of the bacterial species here assessed are members of the ESKAPEE group. We also assessed the stability of the antimicrobial potential of the EOs over a seven-day period. One of the objectives of the present research was to identify the EOs that demonstrated both stability and an extensive spectrum of activities, irrespective of the membrane structure of the bacterial target, and thus contribute to the development of new natural therapeutic approaches that can potentially address the emergence of AMR. Furthermore, we looked into the influence of two distinct solid culture media on the stability and degree of antimicrobial activity of EOs against Gram-positive and Gram-negative bacteria. The most commonly used medium for diffusion tests, namely Mueller-Hinton agar, and a nutritionally rich medium such as Brain Heart Infusion agar, were employed with the objective of contributing to the standardisation of the methods for antimicrobial testing of EOs. To the best of our knowledge, this study is one of the few systematic comparisons investigating the antimicrobial effect and long-term stability of such a large number of EOs against different bacterial species, with testing conducted in two different growth conditions.

## 2. Materials and Methods

### 2.1. Gram-Positive and Gram-Negative Bacteria

A total of 17 bacterial strains were included in the study: nine were reference strains and eight were opportunistic and clinical pathogens isolated from clinical and subclinical mastitis in small ruminants. The classification of these bacteria according to Gram staining is reported in Table 1.

### 2.2. Essential Oils

The following 18 EOs, provided by Cruciani Prodotti Crual Srl (Rome, Italy), were used in the study: cinnamon (*Cinnamomum zeylanicum*), bergamot (*Citrus bergamia*), lemon (*Citrus limonum*), cumin (*Cuminum cyminum*), juniper (*Juniperus communis*), lavender (*Lavandula angustifolia*), laurel (*Laurus nobilis*), tea tree (*Melaleuca alternifoglia*), peppermint (*Mentha piperita*), myrtle (*Myrtus communis*), basil (*Ocimum basilicum*), oregano (*Origanum vulgare*), black pepper (*Piper nigrum*), rosemary (*Rosmarinus officinalis*), sage (*Salvia sclarea*), clove (*Syzygium aromaticum*), thyme (*Thymus vulgaris*) and ginger (*Zingiber officinale*). The main characteristics of the EOs reported in Table 2 were not determined by GC-MS measurements, but rather derived from the manufacturer’s reports (if provided) or the extant literature.

### 2.3. The Spot-on-Agar Test

The antimicrobial activity of the 18 EOs against both the Gram-positive and Gram-negative bacteria listed in Table 1 were tested by the agar spot test on two different agar media, i.e., Brain Heart Infusion (BHI; Becton, Dickinson and Company, Franklin Lakes, NJ, USA) and Mueller–Hinton (MH; Laboratorios Conda S.A., Madrid, Spain) broth. Petri dishes with a diameter of 90 mm and filled with approximately 20 mL of agar medium were employed. Briefly, each strain was cultured in BHI and MH broth for 24 h at 37 °C. Then, the microbial cultures were adjusted to a 0.5 McFarland scale and 500 μL of each suspension were spread onto the surface of either BHI- or MH-agar plate, respectively. In order to facilitate optimal growth conditions for *Listeria monocytogenes* and *Streptococcus uberis*, MH broth and agar were supplemented with lysed horse blood (Thermo Fischer Scientific Inc., Waltham, MA, USA). After the inoculum was absorbed into the agar, one microliter of EO was spotted onto the agar plate with a test organism in triplicate. In addition, one microlitre of PBS was spotted as a negative control. In parallel, the diffusion susceptibility test with Oxoid Amoxicillin/Clavulanic Acid 2:1 30 µg Antimicrobial Susceptibility discs (Thermo Fischer Scientific Inc., USA) was used as a positive control. The plates were then incubated at 37 °C and the diameter of each zone of inhibition or halo was measured in centimetres using a caliper after 24 and 48 h, as well as seven days of incubation. An example of agar spot test is shown in Figure 1. Each experiment was conducted in triplicate. The median inhibition halo diameter (hd) and standard deviations were then calculated on triplicates. The antimicrobial activity of EOs was considered very high when hd was greater than or equal to 2.0 cm (hd ≥ 2.0), high when hd was between 1.5 and 2.0 cm (1.5 ≤ hd < 2.0), moderate when hd was between 1.0 and 1.5 cm (1.0 ≤ hd < 1.5), low when hd was between 0.8 and 1.0 cm (0.8 ≤ hd < 1.0), very low when hd was between 0.8 and 0.5 cm (0.5 ≤ hd < 0.8), and null when hd was less than 0.5 (hd < 0.5).

### 2.4. Statistical Analysis

The efficacy of EOs on two agar media was evaluated by analysing both the frequency and the extent of bacterial growth inhibition. Gram-positive and Gram-negative bacteria were considered separately.

The frequency of bacterial growth inhibition of the 18 EOs on two distinct agar media—BHI and MH—was evaluated following a 24-h exposure period. The statistical analysis employed for this evaluation was the chi-square test of independence. An EO was deemed efficacious when the mean hd was measured to be greater than or equal to 0.8 cm (hd ≥ 0.8), thus incorporating very high, high, moderate and low antimicrobial activities. Conversely, a mean hd smaller than 0.8 cm (hd < 0.8 cm) was considered to be null.

The extent of bacterial growth inhibition of the 18 EOs on two distinct agar media was evaluated by the hds and standard deviations (SDs) following 24-h exposure. Two-way ANOVA followed by Sidak’s multiple comparisons test was performed to determine the significance of the main effects of EO exposure (E) and medium (M), as well as of the interaction between them (E × M). One-way ANOVA followed by a post hoc Tukey’s multiple comparisons test were employed to ascertain the significance of the main effect of EO exposure.

The statistical analyses were conducted using GraphPad Prism 9 version 9.0.0 for Windows, GraphPad Software, La Jolla, CA, USA (www.graphpad.com). Differences were considered significant at a *p*-value of less than 0.05. (*p* < 0.05).

## 3. Results

All bacterial strains exhibited the capacity to proliferate in both BHI and MH media. However, following a 24-h incubation period, a general increase in turbidity (approximately one log increase) was observed in BHI cultures for all tested bacteria. Indeed, an increase in microbial growth was observed, from about 1.0 × 10^8^ CFU/mL in MH to 1.0 × 10^9^ CFU/mL in BHI.

### 3.1. Efficacy of EOs on the Two Growth Agar Media

The frequency of bacterial growth inhibition of the 18 EOs as a whole on BHI and MH after 24 h of incubation is shown in the stacked-bar charts of Figure 2.

The statistical analysis showed a significant difference in EO efficacy by agar medium towards Gram-positive bacteria, X^2^ (1, N = 234) = 3.408, *p* = 0.032. The BHI agar medium was more effective in detecting the antimicrobial activity of EOs (23.4%) in comparison to MH agar (16.7%) after a 24-h incubation period as displayed in Figure 2 (Panel A). Specifically, the red sub-bars (denoting pronounced EO efficacy) decreased from 3.4% to 0.4%, and the magenta sub-bars (indicating substantial EO efficacy) declined from 5.1% to 2.6%. The green sub-bars (indicating moderate EO efficacy) remained consistent across both media, while the yellow sub-bars (indicating low EO efficacy) decreased from 5.1% to 4.3%. Conversely, the white sub-bars (denoting null or very low activity) increased from 76.6% on BHI to 83.3% on MH (Figure 2A).

By contrast, no significant difference in EO efficacy by agar medium was observed towards Gram-negative bacteria, X^2^ (1, N = 72) = 0.046, *p* = 0.415. The overall efficacy of EOs against Gram-negative bacteria appeared to be comparable on the two agar media: 18.1% on BHI and 19.4% on MH (Figure 2B). However, differences were recorded. Specifically, 81.9% of null or very low activities (white sub-bars) were recorded on BHI plates, compared to 80.6% on MH plates. An increase in moderate (green sub-bars) and low (yellow sub-bars) EO efficacies was recorded on MH compared to BHI. However, pronounced EO efficacy was only recorded on BHI (1.4%, red sub-bars), and a higher level of substantial EO efficacy (magenta sub-bars) was recorded on BHI agar plates (6.9%) than on MH agar plates (4.2%) (Figure 2B).

The analysis of the results of the spot-on-agar tests obtained after the three time points considered—namely, at 24 h, 48 h and 7 days—produced congruent results to those displayed at 24 h only. Please, see Appendix A.

The exent of bacterial growth inhibition of the 18 EOs by mean hd measurements on BHI and MH after 24 h of exposure is show in Table 3.

Both the agar medium and the EO exposure affected the mean hd measurements for Gram-positive bacteria and their interaction (E × M) was significant (Table 3). Significant mean hd increases were recorded for oregano (*p* = 0.0006) and thyme (*p* = 0.0194) when Gram-positive bacteria were tested on BHI in comparison with MH, as shown in Figure 3. On the contrary, only the EO exposure significantly affected the mean hd measuremnts in Gram-negative bacteria (Table 3). The results of the main effects of EO exposure on mean hd changes in Gram-positive and Gram-negative bacteria are shown in the statistical analysis reports, Appendix A.

### 3.2. Efficacy of EOs Against Gram-Positive Bacteria on BHI Growth Agar Medium

The results of the antimicrobial properties of 18 EOs against 13 Gram-positive bacteria on BHI agar are detailed in Appendix A.

The overall efficacy of the EOs remained consistent across a period of seven days. The most efficacious EOs against Gram-positive bacteria were oregano, followed by thyme and tea tree (Figure 4A). Oregano demonstrated very high levels of efficacy (hd ≥ 2.0) against *S. aureus* LMG, *S. epidermidis* SIC-14, *S. uberis* LMG and *S. agalactiae* SIC-12. Furthermore, it exhibited high efficacy (1.5 ≤ hd < 2.0) against *E. faecalis* ATCC, *S. aureus* ATCC (but limited to 24 h), *S. dysgalactiae* SIC-10, *S. chromogenes* SAR-15, *S. epidermidis* SAR-16, *S. agalactiae* LMG and *S. intermedius* SIC-8. Oregano showed moderate effects (1.0 ≤ hd < 1.5) against *L. monocytogenes* ATCC and *S. aureus* SIC-11. Conversely, thyme exhibited very high efficacy against *S. epidermidis* SIC-14, *S. agalactiae* SIC-12 and *S. aureus* LMG, albeit for a duration of 24 h, and high efficacy against *S. agalactiae* LMG and *L. monocytogenes* ATCC (limited to 24 h) (Appendix A). Thyme demonstrated moderate effects against all the other Gram-positive bacteria with the exception of *S. aureus* SIC-11, towards which the effect was marginal (0.8 ≤ hd < 1.0). Tea tree exhibited high efficacy against *S. uberis* LMG, *S. agalactiae* LMG and *S. aureus* ATCC (albeit limited to 24 h), and moderate efficacy against *S. aureus* LMG, *S. dysgalactiae* SIC-10, *S. chromogenes* SAR-15 and *S. epidermidis* SIC-14. In contrast, the investigation revealed that tea tree demonstrated minimal or no impact (hd < 0.8) on the remining Gram-positive bacteria. Clove exhibited moderate efficacy against *L. monocytogenes*, *S. aureus* ATCC (limited to 24 h) and *S. aureus* LMG (limited to 48 h). Conversely, the clove EO had marginal or minimal activity (hd < 1.0) against the remaining Gram-positive bacteria (Appendix A).

The most ineffective EOs against Gram-positive bacteria were bergamot, lemon, cumin, myrtle, basil, black pepper, rosemary and ginger, which demonstrated no or minimal activity (hd < 0.8) against the Gram-positive bacteria in this study (Figure 4A).

The most resistant Gram-positive bacteria to the tested EOs were *S. aureus* SIC-11 and *E. faecalis* ATCC, as evidenced by the fact that two out of 18 EOs exhibited antimicrobial properties. Oregano and thyme exhibited moderate (1.0 ≤ hd < 1.5) and marginal (0.8 ≤ hd < 1.0) efficacy against *S. aureus* SIC-11, respectively. Conversely, oregano and thyme were highly and moderately efficacious (hd ≥ 1.0) against *E. faecalis* ATCC, respectively (Appendix A). The most susceptible bacteria were *S. epidermidis* SIC-14 with 6 out of 18 EOs being efficacious (hd ≥1.0), followed by *S. agalactiae* SIC-12 with 5 out of 18 EOs proving efficacious (Appendix A).

It was observed that the efficacy of the 18 EOs differed within the same species. The most significant differences were observed between the two strains of *S. agalactiae*, namely LMG and SIC-12: tea tree was highly efficacious (1.5 ≤ hd < 2.0) towards strain LMG but ineffective against SIC-12; juniper, peppermint and sage were moderately efficacious (1.0 ≤ hd < 1.5) against SIC-12 but no activity was recorded against LMG. Furthermore, divergences were observed also between the two strains of *S. epidermidis*, namely SIC-14 and SAR-16: peppermint, lavender, laurel and tea tree exhibited high or moderate activities (hd ≥ 1.0) against SIC-14, yet null or negligible activities (hd < 0.8) towards SAR-16. A comparison of the antimicrobial activities of EOs of *S. aureus* strains ATCC and LMG revealed similarities, yet distinctions emerged when these profiles were contrasted with that of *S. aureus* SIC-11. It is noteworthy that SIC-11 exhibited an enhanced resistance to EOs compared to both the ATCC and LMG strains (Appendix A).

### 3.3. Efficacy of EOs Against Gram-Positive Bacteria on MH Growth Agar Medium

The results of the antimicrobial properties of 18 EOs against 13 Gram-positive bacteria on MH agar are detailed in Appendix A.

The overall efficacy of the EOs was maintained over the course of a seven-day period. Oregano was found to be the most efficacious EO also in the MH agar experiments, with thyme and tea tree ranking second and third, respectively, as observed in BHI (Figure 4B). However, oregano exhibited a very high level of efficacy (hd ≥ 2.0) against *S. epidermidis* SIC-14 only; high efficacy (1.5 ≤ hd < 2.0) was recorded against *S. aureus* SIC-11, *S. dysgalactiae* SIC-10 (limited to 48 h) and *S. aureus* ATCC (limited to 24 h). Oregano demonstrated moderate effects (1.0 ≤ hd < 1.5) against the remaining Gram-positive bacteria, with the exception of *L. monocitogenes* ATCC, *S. agalactiae* SIC-12 and *E. faecalis* ATCC, towards which oregano showed low or very low antimicrobial activity (hd < 1.0) (Appendix A). Thyme displayed high efficacy against *S. epidermidis* SIC-14 and *S. agalactiae* LMG, and moderate efficacy against *S. aureus* LMG, *S. dysgalactiae* SIC-10, and *S. epidermidis* SAR-16. In addition, moderate efficacy of thyme was observed also against *S. aureus* ATCC and *S. uberis* LMG, but its effectiveness was limited to 24 h; low and very low impacts (hd < 1.0) were detected towards the remaining Gram-positive bacteria. Tea tree displayed high efficacy towards *S. chromogenes* SAR-15 and moderate effects against *S. aureus* LMG, *S. epidermidis* SIC-14, *S. uberis* LMG and *S. agalactiae* LMG; its effectiveness towards the remaining Gram-positive bacteria was found to be less pronounced (hd < 1.0) (Appendix A).

The investigation revealed that bergamot, lemon, cumin laurel, myrtle, basil, rosemary, sage and ginger exhibited insignificant or no activity against Gram-positive bacteria when tested on MH agar (Figure 4B), as observed on BHI.

The most resistant Gram-positive bacteria to the tested EOs on MH agar were *L. monocitogenes* ATCC and *E. faecalis* ATCC, with only one out of 18 EOs being marginally (hd < 1) or moderately (1.0 ≤ hd < 1.5) effective, respectively. The higher resistance of *E. faecalis* to EOs was also observed on BHI. Conversely, the most susceptible bacteria were *S. epidermidis* SIC-14, followed by *S. aureus* SIC-11, *S. chromogenes* SAR-15, *S. agalactiae* LMG and *S. aureus* ATCC, with 3 out of 18 EOs being efficacious (hd ≥ 1.0) (Appendix A). A greater number of bacteria appeared to be susceptible to the 18 EOs on MH than those observed on BHI. *S. epidermidis* SIC-14 was confirmed as the most susceptible strain on both agar media (Appendix A).

Differences in EO sensitivity were also found within species when tested on MH agar. The most significant discrepancies in the antimicrobial profiles were identified for the *S. agalactiae* strains, as observed on BHI. The LMG strain exhibited a higher degree of susceptibility to thyme and tea tree than the SIC-12 strain; conversely, SIC-12 demonstrated a higher degree of sensitivity to juniper and black pepper in comparison with the LMG strain. In contrast to the findings on BHI, discrepancies were identified between the three *S. aureus* strains: clove, peppermint, tea tree and juniper elicited moderate (1.0 ≤ hd < 1.5), low (0.8 ≤ hd < 1.0) or negligible (hd < 0.8) responses within the species. Furthermore, the *S. epidermidis* strains displayed different responses to the antimicrobial activity of EOs, with tea tree being efficacious (hd ≥ 1.0) towards only the SIC-14 strain (Appendix A).

### 3.4. Efficacy of EOs Against Gram-Negative Bacteria on BHI Growth Agar Medium

The results of the antimicrobial properties of 18 EOs against four Gram-negative bacteria on BHI agar are detailed in Appendix A.

Oregano was identified as the most effective EO in combating Gram-negative bacteria, exhibiting a substantial impact against the tested bacteria (hd ≥ 1.5), especially after 24 h of incubation (Figure 5A). *E. coli* SIC-9 exhibited the greatest sensitivity to oregano. Thyme and tea tree were also effective against the bacteria, but to a lesser extent (Figure 5A). Thyme and tea tree both exhibited high (1.5 ≤ hd < 2.0) and low (0.8 ≤ hd < 1.0) effects against *E. coli* ATCC and *S. enterica* ATCC, respectively. Moreover, thyme demonstrated a moderate impact (1.0 ≤ hd < 1.5) towards *E. coli* SIC-9 (Appendix A).

Bergamot, lemon, cumin, juniper, laurel, myrtle, basil, black pepper, rosemary, sage and ginger exhibited negligible (hd < 0.8) activity against Gram-negative bacteria (Figure 5A). The EOs of lavender, peppermint and clove were moderately effective (1.0 ≤ hd < 1.5) against *E. coli* ATCC only (Appendix A).

The most resistant Gram-negative bacterium to the tested EOs was *S. enterica* NCTC, with one out of 18 EOs proving effective. In contrast, the most susceptible bacterium was *E. coli* ATCC, with six out of 18 EOs being efficacious (hd ≥ 1.0) (Appendix A).

A comparable degree of antimicrobial susceptibility to EOs was observed between the Salmonella strains. In contrast, substantial divergences were observed between the two *E. coli* strains: the ATCC strain demonstrated a higher degree of susceptibility to EOs in comparison with the SIC-9 strain (Appendix A).

### 3.5. Efficacy of EOs Against Gram-Negative Bacteria on MH Growth Agar Medium

The results of the antimicrobial properties of 18 EOs against four Gram-negative bacteria on MH agar are detailed in Appendix A.

Oregano was identified as the most efficacious EO, exhibiting high (1.5 ≤ hd < 2.0) or moderate (1.0 ≤ hd < 1.5) activity against all the tested bacteria, followed by tea tree and thyme, which demonstrated moderate to marginal activity against Gram-negative bacteria (Figure 5B).

Bergamot, lemon, cumin, juniper, lavender, laurel, peppermint, myrtle, basil, black pepper, rosemary, sage, clove and ginger exhibited negligible activity (hd < 0.8) against Gram-negative bacteria (Figure 5B). The effect of cinnamon was marginal (hd < 1.0) towards *S. enterica* NCTC only (Appendix A).

The most resistant Gram-negative bacterium to the tested EOs was *S. enterica* ATCC, with one out of 18 EOs proving effective (hd ≥ 1.0). Conversely, the *E. coli* strains exhibited the highest susceptibility, with three out of 18 EOs being efficacious (hd ≥ 1.0) (Appendix A).

Comparable susceptibility profiles to EOs were observed within the Gram-negative species analysed here on MH agar (Appendix A).

## 4. Discussion

We assessed in vitro the antimicrobial activity of 18 EOs against Gram-positive and Gram-negative bacteria by employing the spot-on-agar test on two different agar media across a period of seven days. This approach was adopted in order to identify the most effective and stable EOs with broad-spectrum activities, irrespective of the bacterial membrane structure.

No internationally accepted, standardized methods for assessing natural substances such as EOs have been established so far. The most common methods used for research on EOs derive from those used for testing the antimicrobial activity of antibiotics against microorganisms, and include both quantitative tests such as dilution methods in broth and agar media for the determination of MIC [13,14,15,23], and qualitative tests such as agar diffusion methods for a preliminary screening of antimicrobial potential [13,14,17,18,23]. Although the results obtained by using the diffusion methods are considered less comparable between studies than MICs, because of variables such as the agar medium types, agar thickness, bacterial inoculum and amount of EO used, the antimicrobial potential of EOs based on the size of the inhibition zone is preferred by researchers for screening purposes [13]. In this study, the spot-on-agar was employed as agar diffusion method to assess the antimicrobial activity of 18 EOs against 17 bacterial strains, due to its rapidity, ease of execution and cost-effectiveness. In the near future, MIC determinations of the most effective EOs against the 17 bacterial strains are to be undertaken by our group.

In the absence of guidelines for testing the antimicrobial activity of EOs in vitro, the majority of studies refer to the Clinical Laboratory Standards Institute (CLSI) and the European Committee on Antimicrobial Susceptibility Testing (EUCAST) guidelines for EO testing. Mueller–Hinton agar, the medium of choice for assessing the susceptibility of bacteria to antibiotics, is typically employed to evaluate the antimicrobial properties of EOs by diffusion methods [24,25,26]. However, MH has been shown to be a poor nutrient medium, and as such, it requires supplementation to enable the growth of certain fastidious microorganisms. This may pose a significant challenge when a comparison of various bacterial species is performed [13]. Nutrient agar [27], tryptone soy (TS) agar [17,18], Luria–Bertani agar [28,29,30] and BHI agar [31] are rarely used. In this study, the widely employed MH agar and the highly nutritious BHI agar were used to assess the antimicrobial activity of 18 EOs using the spot-on-agar method. To the best of our knowledge, no evaluation has been performed so far regarding the long-term effects of a wide selection of EOs and bacterial strains on different agar media.

It was observed that the overall antimicrobial activity of the 18 EOs was significantly more pronounced on BHI agar than on MH agar against Gram-positive bacteria. This finding was substantiated by a comparison of the frequency of efficacious growth inhibitions of EOs and their extent on two media. The significant medium-EO exposure interaction recorded for Gram-positive bacteria suggests that the hd changes following EO exposure were different between BHI and MH, with significant higher inhibition haloes for oregano and thyme on BHI. Conversely, no disparities in the efficacy of EO were observed between the two media for the four Gram-negative bacteria. It can be hypothesised that the assessment of a greater number of Gram-negative bacterial strains may yield divergent results on the two agar media. As demonstrated in a recent study, the selection of medium for EO testing may have a significant impact on the determination of the MICs of EOs against pathogenic bacteria [32]. Hulankova tested the efficacy of oregano and cinnamon in three distinct liquid media—MH, TS and BHI—by employing the broth microdilution method. The findings of that study indicated that the mean MIC values of oregano were significantly higher in MH broth compared to BHI or TS broth, suggesting a potential influence of the composition of the culture medium on the magnitude of the antimicrobial activity exhibited by oregano [32]. It has been hypothesised that proteins may bind to active compounds present in EOs and reduce their availability [17,33,34]. Mueller–Hinton contains a higher amount of proteinous components (317.5 g/L) and starch (1.5 g/L) than BHI (28 g/L) and TS (20 g/L). This may explain the lower efficacy of oregano observed by Hulankova [32] in MH. Furthermore, it is not only the quantity of proteinous substances that may be implicated in the interaction, but also their sequence and structure. While bovine serum albumin has been shown to interfere negatively with EO compounds [35], the addition of meat extract to TS broth has been demonstrated to increase the efficacy of EOs [17]. In addition to the quantity, the sequence and the structure of proteinous components contained in the culture medium, as well as the EO itself can play a role. In the study conducted by Hulankova [32], while oregano was found to be less effective in MH broth, cinnamon was not found to be less effective in MH broth [32]. It is conceivable that the components of an EO may diffuse differently into a culture medium, depending on their polarity. The EOs that are richer in hydrophilic compounds can better diffuse into water, while those that are richer in hydrophobic compounds can bind differently to the medium components (starch, proteins, carbohydrates), which, in turn, can affect their antimicrobial activity. Despite a paucity of research in this area, it is hypothesised that the efficacy of an EO on solid media may be influenced by the same factors which play a role in liquid media, such as the medium components and their polarity. In this study the overall antimicrobial activity of EOs was found to be less pronounced in solid MH than BHI media. This suggests that the medium composition may have had a leading part in the observed interactions. It is worthy of note, however, that the growth medium did not affect the order of effectiveness of EOs against bacteria. Regardless of the solid medium used, oregano was the most effective against both Gram-positive and Gram-negative bacteria, followed by thyme and tea tree. Bergamot, ginger, lemon, cumin, laurel, myrtle, basil, black pepper and rosemary, on the other hand, were inactive or presented minimal antimicrobial activity against both types of bacteria. It is important to note that the overall efficacy of the EOs remained consistent across a period of seven days.

The antimicrobial efficacy of an EO against bacteria can be attributed to its diverse bioactive constituents and their specific mode of action. Despite the absence of characterisation of the compositions of the EOs here used via GC-MS measurements, reference was made to the reports of the manufacturer and relevant literature for the identification of the primary bioactive components that could provide a rationale for the antimicrobial properties observed in this study. Thymol and carvacrol, a phenolic isomer of thymol, are the main bioactive compounds of both thyme and oregano EOs. They are considered key compounds of the monoterpene group and have been extensively studied for their antibacterial properties against both Gram-positive and Gram-negative bacteria [36,37]. Terpinen-4-ol, a monoterpene alcohol, has been identified as the primary bioactive compound of the tea tree EO. Monoterpene alcohols and phenolic derivatives have been identified as some of the most potent antibacterial agents. The antimicrobial properties of these compounds are closely related to their physiochemical properties [38] and are attributable to a cascade of reactions involving the entire bacterial cell [39]. The predominant mechanisms of antibacterial action for these monoterpenes is the structural and functional alterations in the cell membrane and disruption of cell wall integrity, resulting in bacterial cell death [11,40,41,42]. It is evident that there are several additional mechanisms which must be considered, including the induction of reactive oxygen species (ROS) accumulation and DNA damage, the inhibition of efflux pumps, and the disruption of transcriptional pathways [10,39,43,44].

In general, Gram-negative bacteria demonstrate a higher level of resistance to EOs than Gram-positive bacteria [45,46]. Our research found that oregano, thyme and tea tree exhibited the most effectiveness against both Gram-positive and Gram-negative bacteria, despite their structural differences in the cell wall. Gram-positive bacteria are characterised by a thick peptidoglycan layer in their cell wall. This layer facilitates the penetration of hydrophobic molecules, such as EOs compounds, into the cells. These molecules can then act on various components of the bacterial cell, including the cell wall, cell membrane and cytoplasm [39]. In contrast, Gram-negative bacteria possess a thin peptidoglycan layer that is coated by an outer membrane. The outer membrane is composed of a double layer of phospholipids, which are linked to the cell membrane by lipopolysaccharides. The outer membrane displays a low degree of permeability to hydrophobic molecules, thus rendering Gram-negative bacteria more resistant to EOs than Gram-positive bacteria [39]. However, thymol and carvacrol have been observed to disrupt outer membranes through two distinct mechanisms: the release of lipopolysaccharides [47,48,49] and the alteration of the folding and insertion of outer membrane proteins [50]. In addition, there is evidence that thymol and carvacrol can act on cell membranes by altering their fatty acid composition. This, in turn, can increase the fluidity and permeability of the cell membrane, resulting in cell lysis and death [51,52,53,54]. Greater effectiveness of oregano, thyme and tea tree among the EOs here tested can be explained by their ability to penetrate the cell walls of Gram-negative and Gram-positive bacteria and affect the integrity of their cell membranes. The superior performance of oregano and thyme among the tested EOs was also documented against the Gram-negative bacterium *Acinetobacter baumannii* [42], as well as against other Gram-strain bacterial species [45]. Concordant results with the literature were also observed for the less effective EOs. The negligible activity of lemon against *S. aureus* and *E. coli* has also been documented by [55]. Gutierrez et al. [17] found that rosemary, basil, lemon and sage were less effective than oregano and thyme against *E. coli* and *L. monocytogenes*.

The cell membrane constitutes the primary target of EOs, yet this is not the sole mechanism by which EOs exert their effects. Wang et al. [54] discovered that thymol, once inside the cell, is able to bind to the minor groove of the DNA, leading to the slight destabilization of the DNA’s secondary structure. Furthermore, thymol has been demonstrated to disrupt energy metabolism, nucleotide biosynthesis and DNA repair processes [10,56]. Therefore, the antimicrobial activity of an EO is not due to a single mode of action, but involves several targets in the cell [45].

Discrepancies in the composition and structure of the cell wall, cell membrane, cytoplasm, and other intracellular targets of EOs can therefore not only elucidate why Gram-negative bacteria demonstrate greater resistance to EOs than Gram-positive bacteria, but also explain the variations in EO effects documented among strains of the same bacterial species. Here, we document differences in EO susceptibility also between strains belonging to the same species, in both the Gram-positive (*S. agalactiae*, *S. epidermidis* and *S. aureus*) and Gram-negative (*E. coli*) groups. Intraspecies differences are not limited to bacteria, as they have also been documented in yeast and mould species [57]. Intrastrain variations in EO susceptibility have also been described, and have been found to be linked to distinct bacterial growth phases [58]. In our study, however, oregano appears to bypass inter- and intraspecies differences more effectively than other EOs, proving effective against all the tested bacteria, albeit to varying degrees. The only exception was *E. faecalis*, against which oregano was ineffective.

## 5. Conclusions

In conclusion, this study represents one of the few systematic comparisons of the antimicrobial effect and long-term stability of such a large number of EOs against different bacterial species, with the testing conducted under two distinct growth conditions. The large variability in the mechanisms of action of an EO against bacterial species and strains, as documented in the literature, can be attributed to multiple factors. These factors are related to the EOs themselves, such as their chemical composition, the method used to test their antimicrobial activity (e.g., incubation time, culture medium, emulsifiers/solvents) and the microorganisms tested. The latter may exhibit different sensitivities to EOs based on their species- and strain-specific characteristics. However, there is a consensus that oregano and thyme are the most effective antibacterial EOs, regardless of the methods and procedures used in antimicrobial susceptibility testing or the species and strain tested. While oregano and thyme have demonstrated potential in the development of novel antimicrobial agents, further research is necessary to elucidate any antagonistic effects or interactions of these EOs and to validate their effectiveness and safety through quantitative and in vivo studies.

## Figures and Tables

**Figure 1 pathogens-14-01216-f001:**
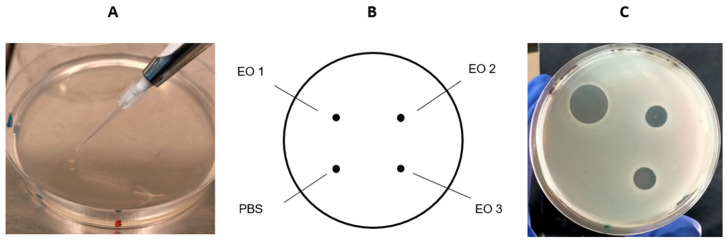
Example of the spot-on-agar test. (**A**) A droplet (one microliter) of the test EO (or PBS) was carefully spotted onto the agar surface previously seeded with a bacterial strain. (**B**) Each plate was spotted with PBS (negative control) and three different EOs. (**C**) The diameter of the inhibition zones (haloes) was measured in cm after incubation.

**Figure 2 pathogens-14-01216-f002:**
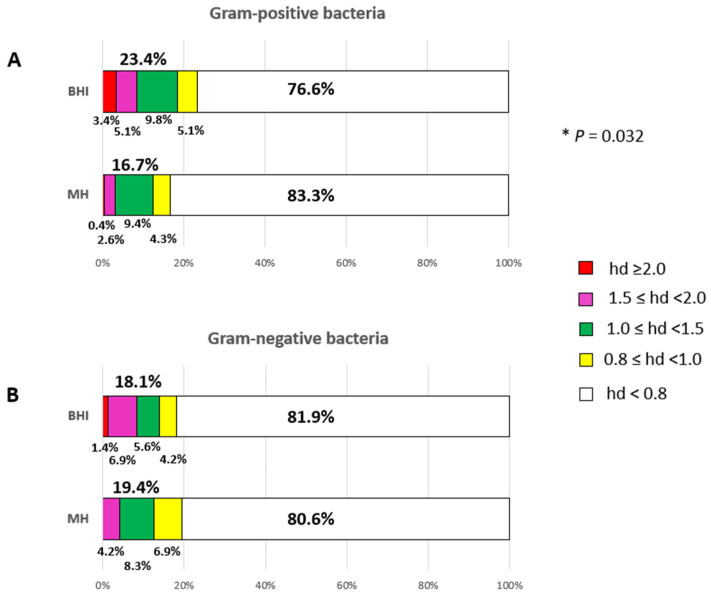
Overall efficacy of EOs by assessing the frequency of efficacious bacterial growth inhibition on BHI and MH agar plates after a 24-h incubation period. Panel (**A**), antimicrobial power of EOs against Gram-positive bacteria. Panel (**B**), antimicrobial power of EOs against Gram-negative bacteria. Sub-bars indicating the frequency of very high (red), high (magenta), moderate (green), low (yellow) and very low/null (white) antimicrobial activities are shown. The efficacy of EOs is measured by the diameter of the inhibition zones (haloes) in centimetres after incubation. * *p* ≤ 0.05.

**Figure 3 pathogens-14-01216-f003:**
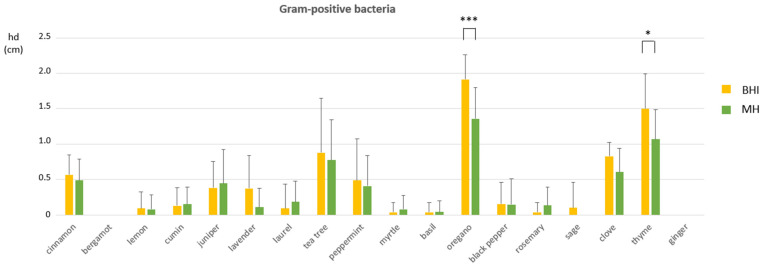
The interaction effect of medium and EO exposure on inhibition halo diameters (hd) among Gram-positive bacteria. Mean hds ± SDs among EOs are shown. Between-group analysis by two-way ANOVA followed by Sidak’s multiple comparisons tests, were performed to determine the significance of the main effects of EO exposure and medium, as well as of the interaction between them. Asterisks indicate statistical significance: * *p* ≤ 0.05 and *** *p* ≤ 0.001.

**Figure 4 pathogens-14-01216-f004:**
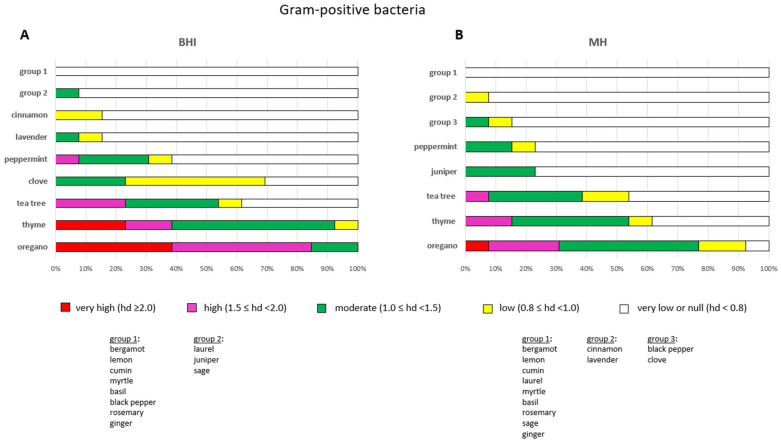
Efficacy of the antimicrobial activity of each EO after a 24-h incubation period towards the 13 Gram-positive bacteria selected in this study. The antimicrobial activity of EOs was tested on BHI (**A**) and MH (**B**). Sub-bars indicating the frequency of very high (red), high (magenta), moderate (green), low (yellow) and very low/null (white) antimicrobial activities are shown. The efficacy of each EO is measured by the diameter of the inhibition zones (hds) in centimetres after incubation.

**Figure 5 pathogens-14-01216-f005:**
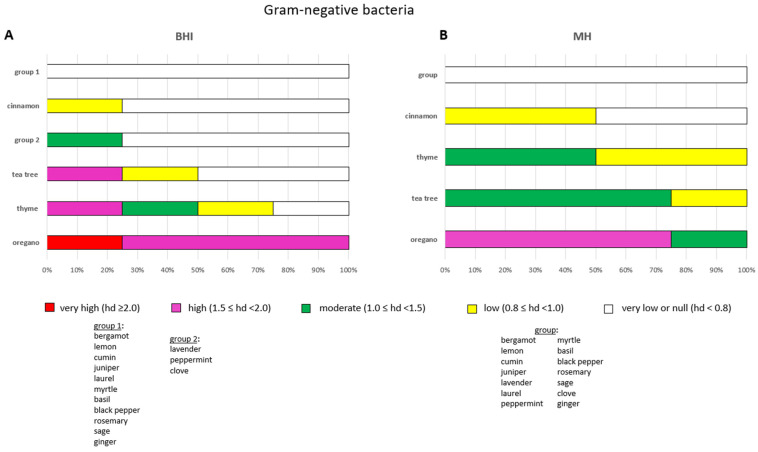
Efficacy of the antimicrobial activity of each EO after a 24-h incubation period towards the four Gram-negative bacteria selected in this study. The antimicrobial activity of EOs was tested on BHI (**A**) and MH (**B**). Sub-bars indicating the frequency of very high (red), high (magenta), moderate (green), low (yellow) and very low/null (white) antimicrobial activities are shown. The efficacy of each EO is measured by the diameter of the inhibition zones (hds) in centimetres after incubation.

**Table 1 pathogens-14-01216-t001:** List of the Gram-positive and Gram-negative bacterial strains used in this study.

Bacteria	Collection Code
Gram-positive strains	
*Enterococcus faecalis*	ATCC 29212 *
*Listeria monocytogenes*	ATCC 19111 *
*Staphylococcus aureus*	ATCC 6538 *
*Staphylococcus aureus*	BCCM/LMG 16805 *
*Staphylococcus aureus*	SIC-11 ^
*Streptococcus dysgalactiae*	SIC-10 ^
*Staphylococcus chromogenes*	SAR-15 ^
*Staphylococcus epidermidis*	SIC-14 ^
*Staphylococcus epidermidis*	SAR-16 ^
*Streptococcus uberis*	BCCM/LMG 14750 *
*Streptococcus agalactiae*	BCCM/LMG 14838 *
*Streptococcus agalactiae*	SIC-12 ^
*Staphylococcus intermedius*	SIC-8 ^
Gram-negative strains	
*Salmonella enterica* subsp. *enterica CDC 6516-60*	ATCC 14028 *
*Salmonella enterica* subsp. *enterica SL1344*	NCTC 13347 *
*Escherichia coli Seattle 1946*	ATCC 25922 *
*Escherichia coli*	SIC-9 ^

* Reference strains; ^ mastitis-causing pathogens from our collection.

**Table 2 pathogens-14-01216-t002:** Essential oils used in the study and their characteristics.

EO	Plant Name (Plant Source) ^	Main Components *
Cinnamon	*Cinnamomum zeylanicum* (leaf)	cinnamic aldehyde (65–80%), eugenol (4–10%)
Bergamot	*Citrus bergamia* (peel)	limonene (30–60%), linalyl acetate (15–25%), linalool (3–10%)
Lemon	*Citrus limonum* (peel)	limonene (70%)
Cumin	*Cuminum cyminum* (seed)	camphene (25–30%), cuminic aldehyde (20–30%)
Juniper	*Juniperus communis* (berry)	α-pinene (27–35%), sabinene (10%), limonene (2–7%) [20]
Lavender	*Lavandula angustifolia* (flower)	linalyl acetate (40%)
Laurel	*Laurus nobilis* (leaf)	cineole (30–50%)
Tea tree	*Melaleuca alternifolia* (leaf)	terpinen-4-olo (40%), *γ*-terpinene (22%)
Peppermint	*Mentha piperita* (leaf)	menthol (30–50%), mentone (20–30%)
Myrtle	*Myrtus communis* (leaf)	myrtenyl acetate (21%), 1,8-cineol (17%), α-pinene (16%), linalool (13%), limonene (9%), linalyl acetate (4%), geranyl acetate (3%), α-terpineol (3%) [21]
Basil	*Ocimum basilicum* (leaf)	methyl chavicol (70–80%), linalool (15–25%)
Oregano	*Origanum vulgare* (leaf and flower)	carvacrol (80–85%), *γ*-terpinene (5–7%), *p*-cymene and thymol [18]
Black pepper	*Piper nigrum* (berry)	monoterpene (70–80%)
Rosemary	*Rosmarinus officinalis* (flower)	cineole (35–45%)
Sage	*Salvia sclarea* (flower)	linalyl acetate (45–70%)
Clove	*Syzygium aromaticum* (bud)	eugenol (85–95%)
Thyme	*Thymus vulgaris* (leaf and flower)	thymol and carvacrol (60%)
Ginger	*Zingiber officinale* (root)	zingiberene (19.71%), (+)-β-cedrene (12.85%), farnesene (12.17%), α-curcumene (10.18%), β-elemene (3.54%) [22]

^ The part of the plant used to produce the EO is indicated in brackets, as specified by the manufacturer. * The chemical composition was obtained from either the manufacturer or relevant literature in the case of missing data.

**Table 3 pathogens-14-01216-t003:** Mean inhibition halo diameters by EO exposure group and agar medium in Gram-positive and Gram-negative bacteria.

EO Exposure Group	Gram-Positive Bacteria			Gram-Negative Bacteria		
	BHI	MH		BHI	MH	
	Mean hd ± SDs	(N = 13)	*p*-Value	Mean hd ± SDs	(N = 4)	*p*-Value
Cinnamon	0.570 ± 0.275	0.489 ± 0.298	E < 0.0001 **** M = 0.0060 ** E × M = 0.0223 *	0.600 ± 0.181	0.595 ± 0.421	E < 0.0001 **** M = 0.2130 E × M = 0.9644
Bergamot	0.0 ± 0.0	0.0 ± 0.0		0.0 ± 0.0	0.0 ± 0.0	
Lemon	0.095 ± 0.234	0.082 ± 0.201		0.0 ± 0.0	0.0 ± 0.0	
Cumin	0.131 ± 0.251	0.154 ± 0.24		0.0 ± 0.0	0.0 ± 0.0	
Juniper	0.382 ± 0.377	0.445 ± 0.478		0.265 ± 0.306	0.0 ± 0.0	
Lavender	0.374 ± 0.464	0.110 ± 0.271		0.275 ± 0.550	0.143 ± 0.285	
Laurel	0.095 ± 0.341	0.187 ± 0.292		0.133 ± 0.265	0.143 ± 0.285	
Tea tree	0.882 ± 0.768	0.774 ± 0.567		0.950 ± 0.542	1.058 ± 0.217	
Peppermint	0.487 ± 0.588	0.41 ± 0.426		0.358 ± 0.715	0.0 ± 0.0	
Myrtle	0.038 ± 0.139	0.079 ± 0.193		0.0 ± 0.0	0.125 ± 0.250	
Basil	0.038 ± 0.139	0.044 ± 0.158		0.0 ± 0.0	0.0 ± 0.0	
Oregano	1.915 ± 0.345	1.358 ± 0.444		1.750 ± 0.311	1.608 ± 0.262	
Black pepper	0.157 ± 0.301	0.148 ± 0.363		0.0 ± 0.0	0.0 ± 0.0	
Rosemary	0.038 ± 0.139	0.134 ± 0.257		0.175 ± 0.350	0.150 ± 0.300	
Sage	0.1 ± 0.361	0.0 ± 0.0		0.0 ± 0.0	0.0 ± 0.0	
Clove	0.831 ± 0.193	0.605 ± 0.332		0.775 ± 0.152	0.683 ± 0.043	
Thyme	1.504 ± 0.484	1.069 ± 0.422		1.148 ± 0.397	0.968 ± 0.124	
Ginger	0.0 ± 0.0	0.0 ± 0.0		0.0 ± 0.0	0.0 ± 0.0	

SD, standard deviation; hd, inhibition halo diameter; N, number of bacteria. *p*-values were calculates using two-way ANOVA followed by Sidak’s multiple comparisons test to determine the significance of the main effects of EO exposure (E) and medium (M), as well as of the interaction between them (E × M). Asterisks indicate statistical significance: * *p* ≤ 0.05, ** *p* ≤ 0.01, and **** *p* ≤ 0.0001.

## Data Availability

All relevant data is contained within the article.

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
