# Peer review of "In Vitro Evaluation of the Antimicrobial Activity of Eighteen Essential Oils Against Gram-Positive and Gram-Negative Bacteria in Two Different Growth Media"

_pathogens, 2025, doi:10.3390/pathogens14121216_

Round 1

Reviewer 1 Report

Comments and Suggestions for Authors

Dear Authors,

Your manuscript entitled “In vitro evaluation of the antimicrobial activity of eighteen essential oils against Gram-positive and Gram-negative bacteria in two different growth media” deals with a topic that is timely.

General comments:

Unfortunately, I could not find a clear statement on why you wish to compare the antimicrobial activity on these two media. As this is one of the main aim of your research this statement have to be made clearly.

The method used is semi-quantitative, appropriate for preliminary results, but not suitable for drawing major conclusions.

Some of the statements made in the abstract and discussion are generalized, especially those about clinical applications or stability, and are not supported by your results.

Point-by-Point comments

Abstract

Line 13–14: Add the word method to make it clear: 'spot-on-agar method using two different culture media.'

Line 15–17: Provide quantitative proof for repeated efficiency claim or reword statement.

Line 20–22: Unexplained assumptions, Avoid or explain how these findings might result in new natural cures, as antimicrobial activity of investigated Eos is well established, or explain how will it result in forming standard protocol.Introduction

Line 43–49: change 'Staphylococcus aures' to 'Staphylococcus aureus.'
Line 74–75: general statement—add citation or qualify.
Line 79–101: Clarify limitations of diffusion vs dilution methods.
Line 115–118: Claim 'first study' is  to brave, change to 'one of the few systematic comparisons’ or similar.'

Materials and Methods

Line 130–143: EO composition must be determined by a GC–MS measurements or acknowledged as a limitation of this study.
Line 145–165: Specify all relevant data for reproducibility of your results such as plate size, measurement method, control type (include positive control), and define 'median halo diameter.'

Results

Line 174–175: statment 'one log increase in turbidity' was turbidity measured? if so provide data, if it is only visualy observed change to 'visible increase in turbidity.'
Line 177–207: Statistical data needed for Figure 2.
Line 220–265: Support claims proper statistical analsys.
Line 267–306: Compare BHI vs MH data directly.
Line 309–347: Explain mechanisam for reduced Gram-negative susceptibility.

Discussion and Conclusion

Avoid exaggeration of in-vitro results as potential treatments.

Address discussion on mechanisms and medium effect.

Add paragraph on study limitations (no MIC/MBC, evaporation, no GC–MS validation).

Conclude modestly: suggest future quantitative and in vivo studies.

 Key points for major revisions

  1. Add MIC/MBC assays for most active essential oils for both media.
  2. Add positive antibiotic control.
  3. Validate EO composition by GC–MS for most active essential oils.
  4. Apply appropriate statistical analyses.
  5. Condense large tables; eliminate pie charts and substitute bar/box plots.

Author Response

We would like to express our gratitude to the Reviewer for their constructive feedback. Modifications are highlighted in yellow in the text of the revised version of the manuscript.

General comments:

1-Unfortunately, I could not find a clear statement on why you wish to compare the antimicrobial activity on these two media. As this is one of the main aim of your research this statement have to be made clearly.

As suggested, we have now provided a more comprehensive clarification in the introduction section that, along with the range of testing methods described in the literature, a variety of media have also been employed for the execution of EO testing (see lines 94-96). It is now evident that the objective of this study was to examine the impact of two distinct solid culture media and to make a contribution to the standardisation of the methods for antimicrobial testing of EOs, as outlined in lines 112-117.

2-The method used is semi-quantitative, appropriate for preliminary results, but not suitable for drawing major conclusions.

As suggested, the section of the conclusion was subjected to rewriting

3-Some of the statements made in the abstract and discussion are generalized, especially those about clinical applications or stability, and are not supported by your results.

Both the abstract and discussion sections were revised accordingly to the results.

Point-by-Point comments

4-Abstract

Line 13–14: Add the word method to make it clear: 'spot-on-agar method using two different culture media.

Line 15–17:. Provide quantitative proof for repeated efficiency claim or reword statement.

Line 20–22: Unexplained assumptions, Avoid or explain how these findings might result in new natural cures, as antimicrobial activity of investigated Eos is well established, or explain how will it result in forming standard protocol.

As suggested, Abstract was revised accordingly.

5-Introduction

Line 43–49: change 'Staphylococcus aures' to 'Staphylococcus aureus.
Line 74–75: general statement—add citation or qualify.
Line 79–101: Clarify limitations of diffusion vs dilution methods.
Line 115–118: Claim 'first study' is  to brave, change to 'one of the few systematic comparisons’ or similar.'

The typographical error has been rectified and reference number 7 has been included. Furthermore, the text clarifies the limitations of diffusion methods in comparison to dilution methods (see lines 86-92) and the claim “first study” was changed as suggested (lines 117-118).

6-Materials and Methods

Line 130–143: EO composition must be determined by a GC–MS measurements or acknowledged as a limitation of this study.
Line 145–165: Specify all relevant data for reproducibility of your results such as plate size, measurement method, control type (include positive control), and define 'median halo diameter.'

The text clarified the absence of GC-MS measurements (lines 140-142). As recommended, additional methodological information was provided in the texts (see lines 153-168).

7-Results

Line 174–175: statment 'one log increase in turbidity' was turbidity measured? if so provide data, if it is only visualy observed change to 'visible increase in turbidity.'
Line 177–207: Statistical data needed for Figure 2.
Line 220–265: Support claims proper statistical analsys.
Line 267–306: Compare BHI vs MH data directly.
Line 309–347: Explain mechanisam for reduced Gram-negative susceptibility.

As previously outlined, the observed rise in turbidity was explained (lines 206-208). Furthermore, a statistical analysis paragraph was incorporated into the Material and Methods section (lines 181-201), and the results were updated to reflect the statistical analysis. New figures comparing BHI and MH results are provided. Potential mechanisms for reduced Gram-negative susceptibility are outlined in Discussion (lines 593-619).

8-Discussion and Conclusion

Avoid exaggeration of in-vitro results as potential treatments.

Address discussion on mechanisms and medium effect.

Add paragraph on study limitations (no MIC/MBC, evaporation, no GC–MS validation).

Conclude modestly: suggest future quantitative and in vivo studies.

The Discussion and Conclusion sections have been revised in accordance with the reviewer's suggestions. As outlined in the text (lines 523 to 524 and 574 to 578), the study's limitations have been thoroughly described. The text refers to future quantitative and in vivo studies.

9-Key points for major revisions

  1. Add MIC/MBC assays for most active essential oils for both media.

The text outlines future quantitative studies in lines 513 and 655.

  1. Add positive antibiotic control.

Please note that the positive control has been included in the Materials and Methods section.

  1. Validate EO composition by GC–MS for most active essential oils.

The GC-MS measurements of the most active EOs will be conducted as part of our next MIC determination study. As outlined on lines 574-578, the absence of GC-MS characterisation is considered a study limitation.

  1. Apply appropriate statistical analyses.

The statistical analysis was incorporated, and the Results and Discussion sections were updated accordingly.

  1. Condense large tables; eliminate pie charts and substitute bar/box plots.

As recommended, the results of the EO efficacy were set out in a synthetic table (see Table 1). All large tables (previously referred to as Tables 1-4) have now been relocated to the supplementary materials. The pie charts were substituted with stacked-bar charts.

Reviewer 2 Report

Comments and Suggestions for Authors

The manuscript deals with a laboratory study evaluating the antimicrobial effects of 18 essential oils on 17 bacterial strains representing both Gram-positive and Gram-negative groups, tested by the spot-on-agar method on BHI and MH media. The topic is relevant and timely, especially in the context of antimicrobial resistance and the increasing interest in natural agents as alternatives to antibiotics.

Strengths

- Wide selection of essential oils and bacterial strains, including clinical and reference isolates.
- Interesting comparison between two solid media (BHI and MH), which could be useful for future methodological standardization.
- The attempt to evaluate the stability of antimicrobial activity during seven days is valuable and not commonly included in similar works.

Weaknesses and Main Remarks

  1. Semi-quantitative character of the method
    The spot-on-agar approach is suitable for screening, but it does not provide quantitative data. Determination of MIC or MBC values for the most active oils (for example oregano, thyme, or tea tree) would give the results stronger scientific meaning.
    2. Lack of statistical analysis
    The authors mention triplicate assays, but there is no statistical evaluation. A simple comparison, e.g. ANOVA or Kruskal–Wallis, would increase the reliability of the findings.
    3. Discussion too descriptive
       Much of the discussion repeats the results. It would be good to focus more on explaining why certain oils are more active, referring to their main compounds (carvacrol, thymol, terpinen-4-ol, etc.) and to structural differences between Gram-positive and Gram-negative cells.
    4. Novelty and added value
       Similar comparative studies have been published earlier. The authors should clearly state what new information this work brings — perhaps the double-medium comparison or long-term stability aspect.
    5. Presentation of data
       Tables are extremely large and hard to read. Condensing the information into a summary figure (heat map or radar chart) would help the reader.
    6. Minor points
       - Some large tables (3–6) could be moved to Supplementary Material.

Suggestions for Improvement

- Provide at least basic statistical comparison of halo diameters to show significance between oils and media.
- Add or cite MIC/MBC data for key oils to validate the spot-on-agar results.
- Expand discussion on the influence of medium composition on diffusion and microbial growth.
- Shorten the introduction (lines 25–78), which currently contains too much general background on AMR.
- Modify the final conclusion – instead of “paving the way for standard protocols”, I would suggest “supporting further efforts toward standardization of EO testing”.

Author Response

We would like to express our gratitude to the Reviewer for their constructive feedback. Modifications are highlighted in yellow in the text of the revised version of the manuscript.

  1. Semi-quantitative character of the method
    The spot-on-agar approach is suitable for screening, but it does not provide quantitative data. Determination of MIC or MBC values for the most active oils (for example oregano, thyme, or tea tree) would give the results stronger scientific meaning.

We agree with the Reviewer's assessment. The MIC values of the most active EOs will be determined in a future study, as outlined in the Discussion (lines 513 and 655).

  1. Lack of statistical analysis
    The authors mention triplicate assays, but there is no statistical evaluation. A simple comparison, e.g. ANOVA or Kruskal–Wallis, would increase the reliability of the findings.

A statistical analysis paragraph was incorporated into the Material and Methods section (lines 181-201), and the results were updated to reflect the statistical analysis. New figures, a synthetic table and new supplementary materials have been provided. We would like to express our sincere gratitude to the Reviewer. The statistical analysis has significantly enhanced the reliability of our findings and has contributed to the improvement of our study.

  1. Discussion too descriptive
    Much of the discussion repeats the results. It would be good to focus more on explaining why certain oils are more active, referring to their main compounds (carvacrol, thymol, terpinen-4-ol, etc.) and to structural differences between Gram-positive and Gram-negative cells.

We concur with the Reviewer's assessment and have removed redundant results from Discussion.

  1. Novelty and added value
    Similar comparative studies have been published earlier. The authors should clearly state what new information this work brings — perhaps the double-medium comparison or long-term stability aspect.

As recommended, the novelty of our study was clarified in the Introduction section (lines 117 to 118), in Discussion (lines 526 to 527) and Conclusion (642 to 644).

  1. Presentation of data
    Tables are extremely large and hard to read. Condensing the information into a summary figure (heat map or radar chart) would help the reader.

We concur with the Reviewer's assessment that the tables (1-4) were challenging to comprehend. We attempted to substitute them with the suggested heat maps or radar charts, but these proved to be even more challenging to interpret and articulate. We have therefore relocated the large tables (previously referred to as Tables 1-4) to the supplementary materials. A new table and new figures comparing results have been provided in order to facilitate the understanding of our findings by readers.

  1. Minor points

   - Some large tables (3–6) could be moved to Supplementary Material.

As suggested, the large Tables 1-4 were moved to the supplementary materials.

Suggestions for Improvement

- Provide at least basic statistical comparison of halo diameters to show significance between oils and media.

A statistical analysis paragraph was incorporated into the Material and Methods section (lines 181-201), and the results were updated to reflect the statistical analysis.

- Add or cite MIC/MBC data for key oils to validate the spot-on-agar results.

The MIC values of the most active EOs will be determined in a future study, to validate the spot-on-agar results, as outlined in the Discussion (lines 513 and 655).

- Expand discussion on the influence of medium composition on diffusion and microbial growth

The Duscussion section has been revised, as recommended. Please, see lines 520-537.

- Shorten the introduction (lines 25–78), which currently contains too much general background on AMR.

As suggested by the reviewer, the general background on AMR was removed from the introduction section.

- Modify the final conclusion – instead of “paving the way for standard protocols”, I would suggest “supporting further efforts toward standardization of EO testing”.

The conclusions were revised in line with the Reviewer's suggestions.

Round 2

Reviewer 1 Report

Comments and Suggestions for Authors

All points have been addressed, resulting in a significant improvement to the manuscript, which is now acceptable for publishing.

Reviewer 2 Report

Comments and Suggestions for Authors

I accept the revisions made.